# How Initial Forest Cover, Site Characteristics and Fire Severity Drive the Dynamics of the Southern Boreal Forest

**Victor Danneyrolles [1],\***, **Osvaldo Valeria [1]**, **Ibrahim Djerboua [1]**, **Sylvie Gauthier [2]** and **Yves Bergeron [1]**

[1] Institut de recherche sur les forêts, Université du Québec en Abitibi-Témiscamingue, Rouyn-Noranda, QC J9X 5E4, Canada; osvaldo.valeria@uqat.ca (O.V.); djerbouaibrahim@yahoo.fr (I.D.); yves.bergeron@uqat.ca (Y.B.)

[2] Natural Resources Canada, Canadian Forest Service, Laurentian Forestry Centre, Québec, QC G1V 4C6, Canada; sylvie.gauthier2@canada.ca

\* Correspondence: victor.danneyrolles@uqat.ca

**Abstract:** Forest fires are a key driver of boreal landscape dynamics and are expected to increase with climate change in the coming decades. A profound understanding of the effects of fire upon boreal forest dynamics is thus critically needed for our ability to manage these ecosystems and conserve their services. In the present study, we investigate the long-term post-fire forest dynamics in the southern boreal forests of western Quebec using historical aerial photographs from the 1930s, alongside with modern aerial photographs from the 1990s. We quantify the changes in forest cover classes (i.e., conifers, mixed and broadleaved) for 16 study sites that were burned between 1940 and 1970. We then analyzed how interactions between pre-fire forest composition, site characteristics and a fire severity weather index (FSWI) affected the probability of changes in forest cover. In the 1930s, half of the cover of sampled sites were coniferous while the other half were broadleaved or mixed. Between the 1930s and the 1990s, 41% of the areas maintained their initial cover while 59% changed. The lowest probability of changes was found with initial coniferous cover and well drained till deposits. Moreover, an important proportion of 1930s broadleaved/mixed cover transitioned to conifers in the 1990s, which was mainly associated with high FSWI and well-drained deposits. Overall, our results highlight a relatively high resistance and resilience of southern boreal coniferous forests to fire, which suggest that future increase in fire frequency may not necessarily result in a drastic loss of conifers.

**Keywords:** air photo interpretation; disturbance ecology; historical ecology; landscape ecology; mixed boreal forests

## 1. Introduction

Fire is a major driver of ecosystem distribution and dynamics worldwide [1]. In the boreal biome, the forest landscape structure, composition and dynamics are closely related to fire regimes [2–5]. Fire regimes are generally thought to be controlled by bioclimatic factors (e.g., temperature, precipitations, wind, drought, fuel characteristics) as well as direct human controls (e.g., ignition, suppression [6]). Future climate change is also expected to increase wildfire frequency in many parts of the world [7–9], the boreal biome included [10,11]. As such, a profound understanding of the effects of fire upon boreal forest dynamics is crucial for our ability to manage and conserve these ecosystems and their services in our era of rapid global changes.

In the southern boreal forests of eastern Canada, forest post-fire dynamics may be influenced by several interacting factors. Some conifer species are very well adapted to fire and it is common to observe the reestablishment of a coniferous cover immediately after a stand-replacing fire. Jack pine (*Pinus banksiana* Lamb.) has serotinous cones which make its germination totally dependent on fire [12,13]. Black spruce (*Picea mariana* (Mill.) Britton, Sterns and Poggenb.) is more generalist; its semi-serotinous cones allow it to establish rapidly after a fire, but its shade tolerance and vegetative reproduction also allow it to regenerate and maintain its dominance for decades after the fire event [13,14]. However, after a stand-replacing fire, it is also common to observe a rapid installation of broadleaved fast-growing, shade-intolerant trembling aspen (*Populus tremuloides* Michx.) or paper birch (*Betula papyrifera* Marshall) through vigorous root suckering, resprouting or seedling [2,5,15]. Conversely, balsam fir (*Abies balsamea* (L.) Mill.) or eastern white cedar (*Thuja occidentalis* L.) are shade-tolerant fire-sensitive species that are more likely to progressively recolonize and become codominant with black spruce through secondary succession [2,3,16,17]. In any event, the presence or absence of these species in the landscape before a fire (i.e., pre-fire composition) is an essential determinant of post-fire dynamics. Moreover, site characteristics are also critical factors that determine forest composition. For example, jack pine and white birch are particularly well suited to recolonize well-drained and coarser surface deposits (e.g., till, sand deposits or rocky outcrops), while aspen is much more competitive on clay deposits [2,15]. Finally, fire severity may also strongly influence the post-fire changes in forest cover. For example, the opening of serotinous cones relies on high temperature associated with fire, but high severity fire may totally burn the seeds and then prevent coniferous regeneration, while favoring the installation of a broadleaved cover [18]. Yet, potential damage to roots and stumps caused by high severity fire may also restrain the vegetative reproduction of aspen or white birch. Accordingly, to understand post-fire forest dynamics, it appears essential to consider the interactions between pre-fire composition, site characteristics and fire severity.

Here we investigate the long-term post-fire forest dynamics in the southern boreal forests of western Quebec. We used historical aerial photographs from the 1930s, alongside with modern aerial photographs from the 1990s, to quantify the changes in forest cover classes (i.e., conifers, mixed and broadleaved) after fires that occurred during the 1940–1970 period. Compared with other studies that have documented post-fire forest dynamics with space-for-time (i.e., forest chrono-sequences [19]) or dendrochronological approaches [2,15], the strength of our study is the integration of pre-fire composition as a central factor controlling the post-fire dynamic trajectories. We thus analyzed how interactions between pre-fire forest composition, site characteristics and a fire severity weather index (FSWI) affected the probability of changes in forest cover. We also analyze the factors that affected the two main dynamic trajectories observed in our study area: from broadleaved/mixed to coniferous cover and from conifers to broadleaved/mixed cover. Finally, we discuss the implications of these results for a potential increased fire frequency in the southern boreal forests.

## 2. Materials and Methods

### 2.1. Study Area

Our study area is located in the eastern Canadian southern boreal forests (Figure 1), which corresponds to the balsam fir–paper birch bioclimatic domain according to the Quebec's forest classification system [20]. For the 1970–2000 period, mean annual temperatures ranged from −2.5 °C to 0 °C and total precipitations and total annual precipitations ranged between 700 mm and 1000 mm. The flat lowlands (<370 m) of the region are dominated by fine clay deposits that were deposited by the Glacial Lake Ojibway [21], and whose resistance to drainage favor the development of abundant organic soils [22] (Figure 1). Contrastingly, most upland areas (>370 m) are characterized by well-drained glacial tills composed of coarser materials (e.g., sand, gravel and stone with some scattered rocky outcrops; Figure 1).

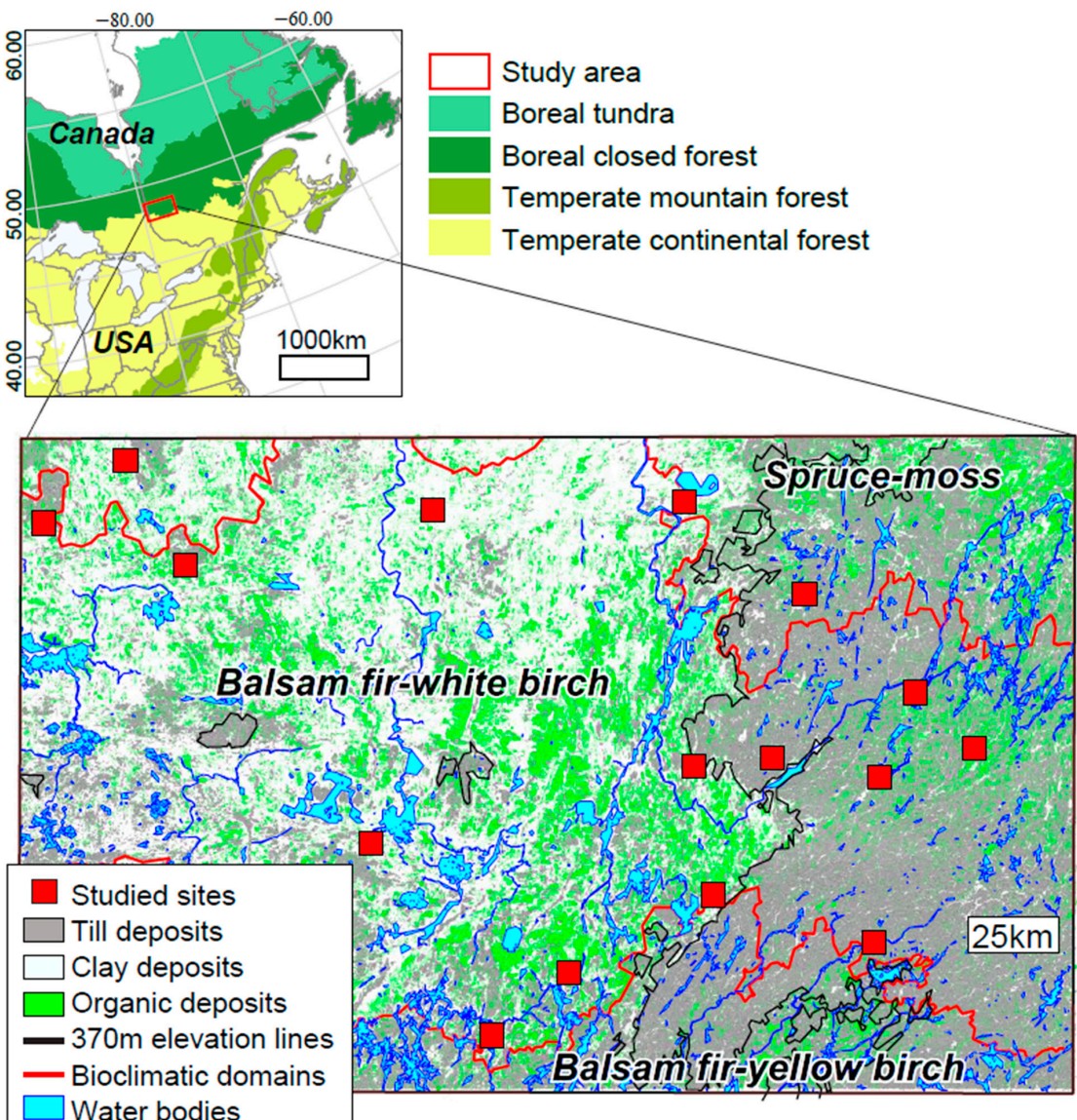

**Figure 1.** Location of the study area in the southern boreal forest of eastern Canada and location of sampled burned sites (red) within the study area. The red lines show the delimitations between bioclimatic domains. Forest zones in the upper panel were modified from the Food and Agriculture Organization of the United Nations (FAO) Global Ecological Zones (GEZ) mapping.

### 2.2. Forest Covers Change and Environmental Data

We documented the post-fire changes in forest cover for 16 study sites (Figure 1) that burned between 1940 and 1970. The 1970 limit was retained in order to avoid young stands difficult to interpret in modern data. Pre-fire forest cover was photo-interpreted with 42 geo-referenced aerial photographs (1:) dating from the 1930–1940 period, available from the National Air Photo Library of Natural Resources Canada. Forest cover categories (conifers, mixed or broadleaved) and stem density classes were photo-interpreted for 226 polygons, with a minimal surface of 4 ha and a mean surface of 12 ha, which summed in a total of 3030 ha. Using a mirror stereoscope, we interpreted the 1930–1940 aerial photographs into visually homogeneous vegetation patches (i.e., polygons) in terms of canopy structure and species composition. This protocol was identical to that used in the Quebec government 3rd decadal forest inventory (1991–2003; photointerpretation of polygons on similar 1: aerial photographs; MFFP 2009). Once delineated, we applied to each polygon a spatial adjustment against a digital terrain model using ArcGIS procedures. Coniferous cover was defined as

cover comprising >75% of coniferous trees, mixed cover between 75% and 25% of coniferous trees, while broadleaved cover was assigned to cover comprising <25% of coniferous trees. Four classes of overall stem density regardless of the species (1: 0–25%, 2: 25–50%, 3: 50–75% and 4: 75–100%) were also identified on 1930s aerial photographs. Prior to analyses, we removed polygons from the 1991–2003 data in which recent secondary disturbances were mentioned (i.e., logging, windthrow, spruce budworm outbreak). Finally, 1930s and 1990s polygons were both rasterized at a 200 × 200 m resolution (4 ha) in order to follow the transition in forest cover for 577 pixels summing 2308 ha.

In order to test the effects of other potential factors that could influence forest dynamics, we collected data about site characteristics and meteorological conditions that could influence fire severity. Site characteristics were documented with three site variables taken from the 1991–2003 forest maps [23]: surface deposits (clay, tills or organic), slope and drainage classes (from very bad: 0, to excessive: 6). Because no direct data on fire severity existed for the 1940–1970 period, fire characteristics were estimated with the meteorological conditions during the month of fire event. Several monthly climate data that are known to directly influence fire severity were obtained with the BIOSIM 10 software (monthly total precipitation, mean temperature, relative humidity and aridity [24]) and used to define two fire severity weather index (FWSI) classes. Fires were considered as potentially severe (high FSWI) when monthly relative humidity was <60% and aridity <15 mm, while other fires were considered as unlikely to be severe (low FSWI).

### 2.3. Statistical Analyses

We first tested the role of different factors upon the overall probability of cover changes, regardless of the direction of changes. Relationships between probability of changes and potential predictors were tested with linear mixed models, which were constructed in R [25] with the *nlme* package [26]. A model selection procedure was constructed in order to evaluate the potential influence of the initial forest cover, site characteristics, FSWI and their interactions (see all variables in Table 1). To avoid collinearity and reduce the dimensions of the explanatory datasets, we first performed a forward selection independently for both three variables categories (i.e., initial forest cover, fire and site characteristics) based on their contribution to the increasing in $R^2$ at each step [27]. We retained the two best variables in each category: initial coniferous cover and tree density classes for the pre-fire forests conditions, fire severity classes and monthly total precipitation during the month of fire for the fire characteristics, and finally drainage classes and till deposits for site characteristics. Then, we tested 11 candidate models that accounted for single sets of variable categories, alongside their possible combinations, with or without interactions (Table 2). In all models, each 16 sampled sites analyzed were used as random factor and within-site spatial autocorrelation was taken into account, using an exponential spatial correlation structure constructed with the *corExp* function contained in the *nlme* package [26]. We also controlled for potential differences in post-fire successional stages between sampled sites by including time since fire (TSF) as a covariable in all models. Time since fire (TSF) represents, for each pixel, the number of years between the 1940–1970 fire event and the date of 1990s photo-interpreted aerial image. Model selection was based on the Akaike information criterion (AIC), in which lowest AIC indicates the best model.

We were also interested in understanding the factors that influence the two directional transitions that summarized the main trends found in our dataset: transition from conifers to mixed or broadleaved cover and transition from mixed or broadleaved to coniferous cover (Figure 2). We used an identical linear mixed model selection procedure as described above, but only using fire and site variables previously retained through forward selection (i.e., fire severity and monthly total precipitation, drainage and till deposits). Candidate models were performed on subsets of the total dataset that comprised only the target initial cover (i.e., conifers or mixed/broadleaved). We then tested four models: (1) fire variables only, (2) site variables only, (3) fire and site variables, and (4) fire and sites variables with interactions.

**Table 1.** Description of explanatory variables.

| Variables | Description |
|---|---|
| | Initial forest characteristics |
| Initial cover | 1930s composition: coniferous, mixed or broadleaved |
| Initial density | 1930s overall stem density: from 1 (low) to 4 (high) |
| | Site characteristics |
| Surface deposits | Three categories: clay, tills or organic |
| Drainage | From very bad (0) to excessive (6) |
| Slope | Slope in % |
| | Monthly fire weather characteristics |
| Total precipitation | Monthly total precipitation in mm |
| Mean temperature | Monthly total precipitation in °C |
| Relative humidity—Aridity | Derived from monthly total precipitation and mean temperature |
| Fire severity weather index | Potentially (high FSWI) or unlikely (low FSWI) severe |

**Table 2.** Competing models of probability of forest cover changes between the 1930s and the 1990s. $P_c$: probability of cover change, $C_{cov}$: pre-fire coniferous cover, Density: pre-fire density cover classes, Till: till surface deposits, Drainage: drainage classes, FSWI: fire severity weather index classes, P: monthly total precipitation during the month of fire and TSF: time since the 1940–1970 fires in the 1990s. The symbol × indicates that interactions were accounted in the models while no interactions were accounted when + symbol is present. The AIC column show the Akaike Information Criterion on which the model selection was based.

| Candidate Models | Variables and Formulas | AIC |
|---|---|---|
| Initial forest | $P_c = C_{cov}$ + Density + TSF | 707.9 |
| Site | $P_c$ = Till + Drainage + TSF | 799.6 |
| Fire | $P_c$ = FSWI + P + TSF | 818.1 |
| Initial forest + Site | $P_c = C_{cov}$ + Density + Till + Drainage + TSF | 692.7 |
| Initial forest + Fire | $P_c = C_{cov}$ + Density + FSWI + P + TSF | 708.2 |
| Site + Fire | $P_c$ = Till + Drainage + FSWI + P + TSF | 820.6 |
| Initial forest + Site + Fire | $P_c = C_{cov}$ + Density + Till + Drainage + FSWI + P + TSF | 699.4 |
| Initial forest × Site | $P_c = (C_{cov}$ + Density) × (Till + Drainage) + TSF | 682.9 |
| Initial forest × Fire | $P_c = (C_{cov}$ + Density) × (FSWI + P) + TSF | 731.1 |
| Init. forest × (Fire + Site) | $P_c = (C_{cov}$ + Density) × (Till + Drainage + FSWI + P) + TSF | 713.0 |

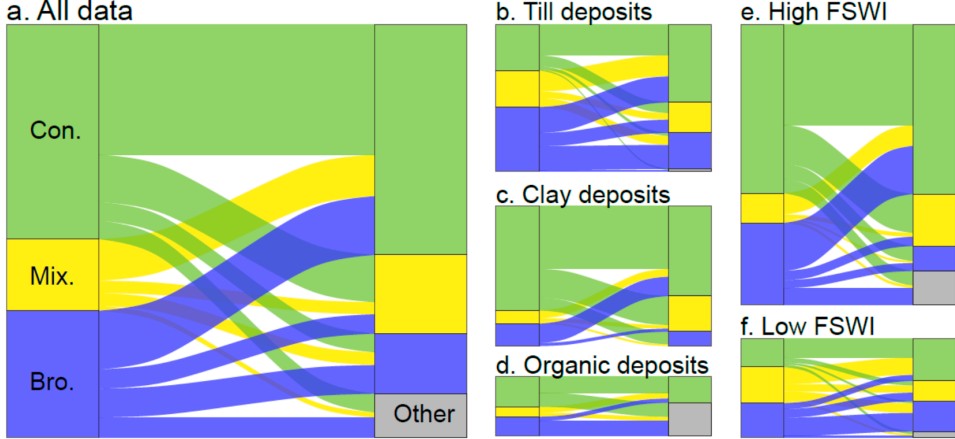

**Figure 2.** Transition diagrams in forest cover classes from 1930s (left) to 1990s (right) for all data (**a**), by deposits classes (**b**–**d**) and by fire severity weather index classes (FSWI; **e**,**f**). Forest classes are coniferous (green), mixed (yellow) and broadleaved (blue) and other non-forested (gray; exclusively wet barrens and alder swamps). The size of individual diagrams in (**b**–**d**) and (**e**,**f**) are proportional to the representation of surface deposits and fire severity classes in the sampled sites.

## 3. Results

In the 1930s, 52% of the cover of the sampled sites was coniferous (Figure 2), while the other half was broadleaved or mixed (31% and 17%, respectively; Figure 2). Coniferous cover was largely dominant on clay deposits (75%; Figure 2), while broadleaved and mixed covers were more common on tills deposits (44% and 25%, respectively; Figure 2). After experiencing one fire in the 1940–1970 period, more than half of the 1930s coniferous covers remained in conifers in the 1990s (61%; Figure 2), while 22%, 9% and 9% switched to mixed, broadleaved and "other" cover classes, respectively (Figure 2). Contrastingly, the majority of 1930s mixed/broadleaved covers switched to other cover classes, mainly to coniferous cover (Figure 2). Finally, cover classified as "others" in the 1990s (exclusively wet barrens and alder swamps; data not shown) accounted for 11% of the initially forested cover in the 1930s.

Overall probability of changes in forest cover classes was firstly driven by initial forest cover and site characteristics: model selection retained the model that accounted for initial composition and density, drainage and till deposits, and their interactions (Table 2). The model explained 44% of the total variance in cover change probability when considering both random and fixed effects, while it explained 30% of the total variance when only considering fixed effects (Figure 3). Pre-fire coniferous cover had the strongest effect on cover change probability, with coniferous cover that had much lower probability of change after fire compared broadleaved or mixed covers. Site drainage also had an important negative effect on cover change probability, and one which significantly interacted with pre-fire cover, with a highly negative effect on coniferous cover while nearly no effect on broadleaved or mixed initial covers (Figure 3). Pre-fire tree density also significantly interacted with site drainage and deposits, with maximum probability of changes for sites with low initial tree density and poorly drained deposits (Figure 3).

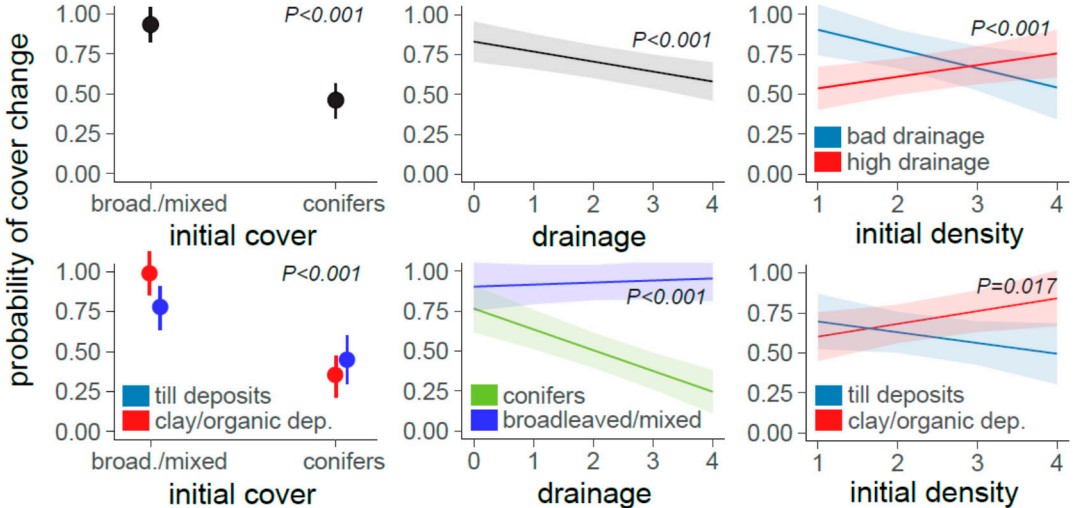

**Figure 3.** Results of overall probability of cover changes modeling. The plots show the predicted values and 95% confidence intervals obtained with the selected linear mixed model that account for initial forest cover, sites characteristics and their interactions ($R^2_{fixed}$ = 0.30; $R^2_{fixed+random}$ = 0.43). Black and white plots show single effects while colored plots show interaction effects. Only significant ($p < 0.05$) effects and interactions are shown (except for forest initial density that had a significant but very weak negative effect). Drainage classes range from bad (0) to good (4), and initial tree cover density ranges from low (1; 0–25%) to high (75–100%).

Probability of change from broadleaved/mixed to coniferous cover was associated with both fire and site characteristics (Figure 4). The model that accounted for FSWI, monthly total precipitation during the month of fire, drainage and surface deposits without interactions was retained with the smallest AIC. This model explained 49% of the variance in the probability of a 1930s broadleaved/mixed cover to change to coniferous when considering both random and fixed effects, while 19% of the

variance when only considering fixed effects. Fire severity and drainage had a significant positive effect, which means that a 1930s broadleaved/mixed cover had higher probability to change to coniferous after a potentially severe fire and on well-drained sites. Conversely, till deposits and time since the 1940–1970s fire (TSF) had negative effects, thus a 1930s broadleaved/mixed cover had higher probability to change to conifers on clay deposits and when fire was most recent.

The results of our analysis of probability of change from a 1930s coniferous to a 1990s broadleaved/mixed cover are not shown since they were largely inconclusive. The model selection retained the model that accounted only for site characteristics (drainage and surface deposits). However, fixed effects model explained only 2% of the variance, with drainage having a significant but very weak positive effect.

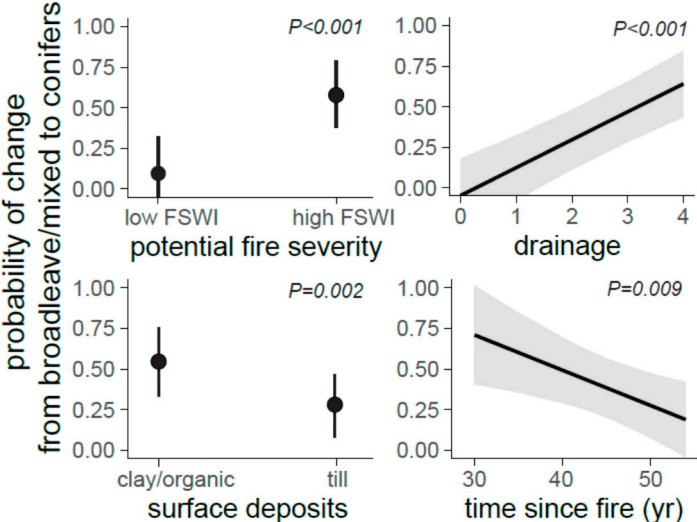

**Figure 4.** Results of modeling of the probability of change from broadleaved/mixed to coniferous cover. The plots show the predicted values and 95% confidence intervals obtained with the selected linear mixed model that account for sites and fire characteristics without interactions ($R^2_{fixed}$ = 0.19; $R^2_{fixed+random}$ = 0.49). Only significant ($p < 0.05$) effects are shown. Drainage classes range from bad (0) to good (4).

## 4. Discussion

The most important aspect that emerges from our results is the high resistance and resilience of southern boreal coniferous forests to fire. Initial coniferous cover had the strongest negative effect upon the overall probability of changes in forest cover, which is likely linked with the ability of jack pine and black spruce to re-establish through abundant seedlings from their serotinous cones after fire [13,18]. This is largely confirmed by the fact that stands that maintained a coniferous cover since the 1930s were dominated by these two species in the 1990s (data not shown from modern maps). The effect of initial coniferous cover also interacted with drainage and surface deposits, with coniferous covers on well-drained tills deposits having a very high probability of maintaining in conifers after fire. This may reflect the fact that jack pine and black spruce mainly rely on mineral substrate to germinate after fire [18,28]. Since well drained till deposits usually have a smaller organic layer thickness compared to clay or organic deposits [29,30], mineral substrates of germination are more likely to be abundant after the passage of fire. On more poorly drained sites, coniferous covers had an important probability to change to broadleaved/mixed on clay sites or to non-forested cover on organic soils. The important proportion of coniferous cover that transited to broadleaved/mixed cover (mostly to poplars dominated stands; data not shown from modern maps) on moderately drained clay deposits (Figure 2) likely reflects the capacity of poplars to vigorously invade the landscape through root suckering after fire on these fine material substrates [2,15,31]. Moreover, compared to well-drained sites that were likely dominated by fire-adapted conifer species (i.e., jack pine and black spruce) before fire, more poorly

drained sites were more likely dominated by fire-sensitive conifer species (i.e., balsam fir and white spruces), which make them more sensitive to a switch to broadleaved/mixed cover. On very badly drained organic deposits, an important proportion of initial coniferous cover changed to non-forested cover (Figure 2; mostly wet barrens) which may be linked to the paludification phenomenon which is common in these lowland areas [22,32,33]. Due to paludification, this is likewise not surprising that badly drained sites with a low initial tree density had a higher probability of change compared to those with high tree density.

A rather surprising result is the important proportion of broadleaved/mixed cover that transited to coniferous cover after a fire event (Figure 2), which seems to be linked to both site and fire characteristics. The broadleaved/mixed covers in 1930 that experienced potentially severe fire on well-drained deposits had the highest probability of changing to coniferous cover. This might reflect that severe fire on these deposits exposes a large amount of mineral substrate that are critical for black spruce and jack pine seedling germination [18], while potentially simultaneously damaging poplars and white birch root systems and then prevent their vegetative reinstallation on these sites. This interpretation still critically depends on the presence of at least some scattered conifer seed trees in the landscape that could recolonize the burnt areas through seed dispersal [34,35]. Nevertheless, the effects of high severity fire may yet prevent the regeneration of conifer species if too large a proportion of cones are burnt. The probability for a mixed/broadleaved cover to change for conifers was also significantly associated with more recent fires (i.e., lowest TSF). This result is in contradiction with the frequently observed postfire successional pathway in which the pioneer broadleaved species (aspen, white birch) are progressively replaced by coniferous species (e.g., black spruce, balsam fir) through succession [2,5]. This may suggest that the important proportion of broadleaved/mixed cover that transited to conifers may be partly linked to a single recent fire event with specific severity characteristics (e.g., high severity).

More generally, the rather unexpected postfire maintenance of conifers and transition from mixed/broadleaved to conifers may also be linked to the particular context of the period on which our analysis focuses. First, all the fires analyzed are from the 1940–1970 period, representing a period when fires were much less frequent compared to the long-term fire history of the study area. In our study area, the transition from the climatically fire-prone Little Ice Age period to the recent warming period (i.e., post 1850) have engendered a transitory decrease in overall burn rates [36–38], but which are currently re-increasing with accelerated climate change [39,40]. As such, the 1940–1970 fires have taken place in a singular climatic period at the regional scale, and which might have altered the postfire dynamics through unusual fire event characteristics or species recruitment and survival. Secondly, forestry practices in our study area generally tend to favor aspen and white birch abundance and may even totally remove conifers seed trees and their aerial seed bank on which they rely for their recolonization after the passage of fire [41,42]. While industrial cutting started in the early 20th century in this region, logging rates remained limited until the apparition of modern mechanized practices the 1960–1970. Thus, the stands that burned during the 1940–1970 period may have only been moderately transformed by forestry practices, which could explain their strong resilience to fire.

## 5. Conclusions

The results of the current study have implications for southern boreal forests management in eastern Canada. As a result of past logging and other human-induced disturbances, the loss of conifers in favor of more mixed and broadleaved landscapes has become important ecological and management issue in the southern boreal forests [31,41–44]. It is also generally considered that projected climate-induced increase in burn rates could likewise exacerbate the loss of conifers in the future [45]. In this context, our results suggest that potential future increase in fire frequency may not necessarily result in a loss of conifers, and in some cases, could even help the reestablishment of a coniferous cover in previously broadleaved or mixed landscapes. Our results highlight that after a fire, the conifer stands on poorly drained sites are the most sensitive to switching to broadleaved/mixed

(for moderately drained clay deposits) or to non-forested wet barrens (for very badly drained sites). As such, these sites should be given special attention in their management in order to not increase their sensitivity in the risk of passage of fire. Conversely, well-drained sites appear to have a better chance to maintain coniferous or even to switch from broadleaved or mixed to conifers. In any event, in the context of a future increased fire activity, the maintenance of a significant abundance of jack pine and black spruce seed trees in the logged stands may greatly benefit the maintenance/reestablishment of a conifers within the landscape.

**Author Contributions:** O.V. and I.B. designed the study. I.D. extracted and compiled the data from historical aerial photographs. V.D. performed the analyses and V.D. and I.D. wrote the manuscript with substantial feedback from O.V., S.G. and Y.B., I.D. and V.D had an equivalent contribution to the manuscript. All authors have read and agreed to the published version of the manuscript.

**Funding:** This project was financially supported by the Natural Sciences and Engineering Research Council of Canada NSERC CRDPJ 394760 – 09.

**Acknowledgments:** We thank Pierre Cartier and Jean Boivin from the CEGEP de l'Abitibi-Témiscamingue, and Carolina Rodriguez from the SIG laboratory of the Université du Québec en Abitibi-Témiscamingue for their help with the photointerpretation and data collection of 1930s aerial photographs. We also thank the Centre d'Information Topographique (Service d'Imagerie Aérienne) of the National Air Photo Library of Natural Resources Canada for their help in locating the studied historical aerial photographs. Finally, this project was financially supported by the Natural Sciences and Engineering Research Council of Canada NSERC CRDPJ 394760 – 09.

**Conflicts of Interest:** The authors declare no conflict of interest.

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
