# Peer review of "How Initial Forest Cover, Site Characteristics and Fire Severity Drive the Dynamics of the Southern Boreal Forest"

_remotesensing, doi:10.3390/rs12233957_

Round 1

Reviewer 1 Report

Dear authors,

The main objective of the research was to study how the initial forest cover, the site characteristics, and the fire severity affect the forest dynamics of the southern boreal forest. This is an interesting question in which remote sensing can greatly contribute, although some ground truth validation would make the results more accurate. I have some general comments/ questions and some specific comments that you may find in the attached pdf.

1) I am not fully convinced with the use of the term “landscapes” in the framework you are using it. May be “study sites” would be better. Otherwise please explain this term selection.

2) The same goes for resinous cover classes. Why not conifers?

3) Not all figures/ results are properly presented in the Results Section. For instance, I would expect to see more information concerning the fire interval (time since fire), which can greatly explain the different succession stages.

3) In the Discussion section you mentioned: “First, all the fires analyzed are from the 1940-1970 period, which exhibits a very low burn rate compared to the long-term and shorter-term regional fire history [33,34].” The reader, especially if he/she is not familiar with forest dynamics and fire history in boreal forests does not understand the meaning of this sentence. I think you should describe/ mention more about the fires in your study sites, their size, frequency, etc. The period of the fire event may have also affected the regeneration pattern. There are some very important aspects that are not properly discussed, especially as far as it concerns the fire events. The Discussion section in general is interesting but quite short and does not cover all your results.

In general, you should be more careful with the terms used in regard to fire ecology (e.g. fire severity, fire regime, etc.).

Author Response

Dear authors,

The main objective of the research was to study how the initial forest cover, the site characteristics, and the fire severity affect the forest dynamics of the southern boreal forest. This is an interesting question in which remote sensing can greatly contribute, although some ground truth validation would make the results more accurate. I have some general comments/ questions and some specific comments that you may find in the attached pdf.

1. I am not fully convinced with the use of the term “landscapes” in the framework you are using it. Maybe “study sites” would be better. Otherwise please explain this term selection. We changed mentions of “landscapes” for “study sites” throughout the entire manuscript.

2. The same goes for resinous cover classes. Why not conifers? We changed “resinous” to “coniferous” or “conifers” throughout the whole text, figures and tables.

3. Not all figures/ results are properly presented in the Results Section. For instance, I would expect to see more information concerning the fire interval (time since fire), which can greatly explain the different succession stages. In our study, Time since fire (TSF) represents, for each study site separately, the number of years between the 1940-1970 fire event and the date of the 1990s photo-interpreted aerial photography. The TSF has been included as a covariable in all models precisely to control for potential differences in succession stages. We clarified this point in the methods section (lines 145-148). Our main analysis did not reveal any significant effect of TSF on the overall probability of cover changes (Figure 3). As such, we didn’t comment or discuss this aspect in the further sections. However, TSF emerged as a significant variable in the analysis of the probability of changes from broadleaved/mixed to resinous cover (Figure 4). As stated in the results section (lines 210-214): “Conversely, till deposits and time since the 1940-1970s fire (TSF) had negative effects, thus a 1930s broadleaved/mixed cover had a higher probability to change to conifers on clay deposits and when the fire was most recent”. We agree that this point was not discussed, and we thus added a few lines in the discussion section (lines 267-272).

4. In the Discussion section you mentioned: “First, all the fires analyzed are from the 1940-1970 period, which exhibits a very low burn rate compared to the long-term and shorter-term regional fire history [33,34].” The reader, especially if he/she is not familiar with forest dynamics and fire history in boreal forests does not understand the meaning of this sentence. I think you should describe/ mention more about the fires in your study sites, their size, frequency, etc. The period of the fire event may have also affected the regeneration pattern. There are some very important aspects that are not properly discussed, especially as far as it concerns the fire events. The Discussion section in general is interesting but quite short and does not cover all your results.We agree that this paragraph lacked context and explanations for the reader to understand our point. We thus clarified and expended this section and we hope that this improves the text in the way expected by the reviewer.   

Specific points:  

Line 19: fire is expected to increase? Or maybe you mean fire intensity, fire severity or frequency of forest fires? Changed for “Forest fires are expected to increase” (line 19).

 Line 21: I do not understand how we can manage the ecosystem services provided by forests. Maybe you mean to preserve or conserve them? Please clarify. Changed to “to manage these ecosystems and conserve their services” (line 22).

Line 25: burned or were burned? Changed to “were burned” (line 25). 

Line 44: maybe better: "the boreal biome included" Changed to “the boreal biome included” (line 44) Line 66: This "too severe fire" is hard to understand.

Consider replace it with "high severity fire" or something similar here and elsewhere. Changed to “high severity fire” here (line 66) and elsewhere (lines 68 & 259). 

Figure 1. Is it necessary to use so huge letters? Consider revising the size of the letters. Text sizes in the legends of Figure 1 have been reduced.

Line 105: consider changing the order. We don’t understand what is wrong here with the cover classes list order.

Line 110: Avoid repetition. We removed the last section of the sentence that was indeed a repetition (line 110).  

Line 123: revise English. We changed “the passage of fire” to “the fire event” (line 123).

Line 147: Maybe explain here which value (lowest) indicates the best model. We added, “in which lowest AIC indicates the best model” at the end of the sentence (line 147). 

Figure 3: Plots in the last row are both colored. We don’t understand this comment since yes, as it is mentioned in the legend caption, colored plots show the interaction effect.  

Lines 214-215: Revise English Changed to: “The model selection retained the model that accounted only for site characteristics (drainage and surface deposits). However, model fixed effects only explained 2% of the variance and only drainage had a significant but very weak positive effect.” (Lines 214-217). 

Line 249: Did you consider the distance from unburned conifer stands? Landscape heterogeneity can greatly affect the regeneration pattern. There are several articles dealing with this and maybe you should consider it. We did not consider the distance to unburned stands in our analysis, which indeed could be a potentially important aspect to account for post-fire regeneration. However, collecting such data about historical fire patterns would represent a very time-consuming task and we think it would go beyond the scope of the present manuscript. 

Line 254: a reference is missing here. Unfortunately, we don’t have a reference here, so we added “potentially” to emphasize that this represents speculation.

Reviewer 2 Report

See attached review document.

Author Response

In this manuscript, the authors describe an effort to investigate postfire forest dynamics and cover transitions in the boreal forests of western Quebec, Canada. They use aerial photography to do this, with a set of photographs from the 1930s and a more modern set from the 1990s. Using these photographs, they assess forest cover types, density, and other factors and use this information to evaluate forest change in a set of sites which burned between 1940 and 1970. They find that a majority of sites experienced some cover change, with an especially large portion of change coming from broadleaf and mixed forest transitioning to resinous conifer forest.

The results presented here are interesting in that they suggest certain types of boreal coniferous forests may be more resilient to fire than previously thought in light of potential climate change projections (e.g., Ali et al. 2012, Kelly et al. 2013).

Overall, the manuscript is well put together, though it could use some minor editing for pluralization and grammar. I think a little bit more detail with regard to the photointerpretation process would help readers. It’s also not clear to me where information regarding when each site burned (TSF) came from – a quick citation or clarification of that in the Methods would be helpful. The statistical analyses are fairly straightforward and well-explained. Specific and general comments are below. In our study, Time since fire (TSF) represents, for each study site separately, the number of years between the 1940-1970 fire event and the date of the 1990s photo-interpreted aerial photography. The TSF has been included as a covariable in all models precisely to control for potential differences in succession stages. We clarified this point in the methods section (lines 145-148).

Specific comments:

Line 56: change “rout” to “root” Done (line 56).

Line 108: change “3d“ to “3rdDone (line 108).

 Lines 107-109: A little more explanation of the photointerpretation might be beneficial here, even though the methods are cited in other work. Even a brief sentence or two might help readers. We added some details and clarifications (lines 109-114). 

Line 132: “constructed in the R” to “constructed in R” Done (line 132).

Line 147: Add “information” – “Akaike information criterion” Done (line 149).

Lines 232-234: Walker et al. (2018) might be another good citation here. Indeed, this is a very pertinent reference which was added line 237.

Lines 241 and 242: Add “species” (or “trees”) after “resinous”. Done (lines 245-246). 

Figure 1: The red line for bioclimatic domains is difficult to see. Perhaps making it thicker or a different color would help. Done (see Figure 1).

Reviewer 3 Report

The manuscript uses photointerpretation to determine the interactive effects of fire and site on stability or changes in forest composition on the multi-decadal time scale.

Line 28 ff.

A small point, but I’m a bit confused about setting “resinous” and “broadleaved” in counterpoint as types of trees. Botanists might say gymnosperms vs. angiosperms, loggers might say hardwoods vs. softwoods, in this case I’d go with “conifers” and “broadleaved” as types of trees under investigation. If I’m just out-of-touch here, disregard the comment.

122-128

How reliable are the estimates of fire characteristics based on montly meteorological conditions? Has that been tested and validated for this habitat? On what basis were those categories of high and low FWSI established? Post-hoc from a review of these data or from past tests or?

Table 2

The AIC of candidate models ranges from 693-820. Is this a large range? How is this to be interpreted?

Figure 4

I strongly question using a line to link the data markers on potential fire severity and surface deposits. These seem to be qualitative categorical variables rather than continuous variables. There can be no line or isotonic process that links those.

Author Response

The manuscript uses photointerpretation to determine the interactive effects of fire and site on stability or changes in forest composition on the multi-decadal time scale.

Line 28 ff.: A small point, but I’m a bit confused about setting “resinous” and “broadleaved” in counterpoint as types of trees. Botanists might say gymnosperms vs. angiosperms, loggers might say hardwoods vs. softwoods, in this case I’d go with “conifers” and “broadleaved” as types of trees under investigation. If I’m just out-of-touch here, disregard the comment. We changed “resinous” to “coniferous” or “conifers” throughout the whole text, figures, and tables.

Line 122-128: How reliable are the estimates of fire characteristics based on montly meteorological conditions? Has that been tested and validated for this habitat? On what basis were those categories of high and low FWSI established? Post-hoc from a review of these data or from past tests or? Unfortunately, our fire weather severity index (FWSI) hasn’t been properly tested, but monthly meteorological-related variables are the only proxy available since no direct data on fire severity nor accurate daily meteorological data exist for the 1940-1970 period. Monthly climate variables are, however, commonly used to model fire activity in eastern North America (see for example Chaste et al. 2018 - Biogeosciences - The pyrogeography of eastern boreal Canada from 1901 to 2012 simulated with the LPJ-LMfire model). We agree that this represents a rather arbitrary choice and that’s why we carefully emphasize on “potentially severe (or un-severe) fires” in the interpretations. In any event, we argue that this choice does not strongly impact our results since the FWSI did not emerge as a very important variable in our analyses. It wasn’t included in the model retained for the main analysis (overall probability of cover change; Figure 3). FWSI was only included and discussed for the secondary analysis (probability of change from broadleaved/mixed to conifers; Figure 4) along with the effects of drainage, surface deposits and time since fire. 

Table 2: The AIC of candidate models ranges from 693-820. Is this a large range? How is this to be interpreted? Yes, this is quite an important range, which basically means that adding or removing variables and interactions strongly impact the predictive power of the models. Anyway, the AIC is there used as a relative metric to select the best model and this seems quite clear that the model presented in the manuscript is the best with an AIC of 682.9, which represents a difference of almost 10 with its closest competitor (i.e., AIC=692.7, see Table 2).

Figure 4: I strongly question using a line to link the data markers on potential fire severity and surface deposits. These seem to be qualitative categorical variables rather than continuous variables. There can be no line or isotonic process that links those. We removed the lines in the plots of factor effects (i.e., initial cover, surface deposits, and potential fire severity) in Figure 3 and Figure 4.

Round 2

Reviewer 1 Report

Dear authors,

The manuscript is really improved from the earlier version. I only have some minor comments/ corrections that you may find below.

Line 20: “climate changes” could be replaces by “climate change”

Line 35: “that future increase in fire regimes”. I have not noticed it in the previous review here, but I have made a relevant comment in the main body of the text. Fire severity, intensity or frequency can increase. Fire regime cannot increase, but it can just change.

Lines 111-118: Something is wrong with these sentences. They have to be revised (English and meaning)

Line 155: “Time since fire (TSF) represent” should be replaced by “Time since fire (TSF) represents”

Line 181: “than a half” could be replaced by “than half”

Lines 186-187: “cover of the sites of the 1930s.” English revision is needed.

Lines 231-232: Instead of “However, model fixed effects only explained 2% of the variance and only drainage had a significant but very weak positive effect.” I would suggest sth like this: “However, fixed effects model explained only 2% of the variance, with drainage having a significant but very weak positive effect.”

Line 270: “after the passage of fire” Please rephrase

Lines 276-277: “This interpretation still critically depends on the presence of at least some scattered resinous conifer seed trees in the landscape.” it would be very nice to continue this sentence with sth like this: “.. that could recolonize the burned areas through seed dispersal” and then add some references. In this case “Moreover” in the next sentence should be replaced by “Nevertheless”.

Lines 284-285: “a specific recent fire event with specific severity characteristics”. These two “specific” are rather vague. What do you mean here?

Line 287: “linked the”. I am not sure, but may be a preposition is missing here.

Line 290: “which represent a period” could be replaced by “representing a period”

Line 291: “of in study area” should be replaced by “of the study area”

Author Response

Dear authors,

The manuscript is really improved from the earlier version. I only have some minor comments/ corrections that you may find below.

Line 20: “climate changes” could be replaces by “climate change”. Done.

Line 35: “that future increase in fire regimes”. I have not noticed it in the previous review here, but I have made a relevant comment in the main body of the text. Fire severity, intensity or frequency can increase. Fire regime cannot increase, but it can just change. This comment was indeed in the previous version of the reviews, sorry we missed it. “Fire regime” was changed for “fire frequency”.

Lines 111-118: Something is wrong with these sentences. They have to be revised (English and meaning). Indeed, we reformulated into three sentences (lines 111-117).  

Line 155: “Time since fire (TSF) represent” should be replaced by “Time since fire (TSF) represents”. Done.

Line 181: “than a half” could be replaced by “than half”. Done.

Lines 186-187: “cover of the sites of the 1930s.” English revision is needed. Changed to “the initially forested cover in the 1930s”.

Lines 231-232: Instead of “However, model fixed effects only explained 2% of the variance and only drainage had a significant but very weak positive effect.” I would suggest sth like this: “However, fixed effects model explained only 2% of the variance, with drainage having a significant but very weak positive effect.” These changes have been made.

Line 270: “after the passage of fire” Please rephrase. Changed to “after a fire event”.

Lines 276-277: “This interpretation still critically depends on the presence of at least some scattered resinous conifer seed trees in the landscape.” it would be very nice to continue this sentence with sth like this: “.. that could recolonize the burned areas through seed dispersal” and then add some references. In this case “Moreover” in the next sentence should be replaced by “Nevertheless”. These changes have been made.

Lines 284-285: “a specific recent fire event with specific severity characteristics”. These two “specific” are rather vague. What do you mean here? The sentence was changed to “This may suggest that the important proportion of broadleaved/mixed cover that transited to conifers may be partly linked to a single recent fire event with specific severity  characteristics (e.g., high severity).”

Line 287: “linked the”. I am not sure, but may be a preposition is missing here. We added “linked to”.

Line 290: “which represent a period” could be replaced by “representing a period”. Done.

Line 291: “of in study area” should be replaced by “of the study area”. Done.